# Synergistic antifungal effects of botanical extracts against *Candida albicans*

Eunjin Cho [1]*, Kenneth Acosta [1°], Joshua Henkin[1°], Rinat Abzalimov[2], Ilya Raskin[1]

**1** Department of Plant Biology, Rutgers University, New Brunswick, New Jersey, United States of America,
**2** Advanced Science Research Center at the Graduate Center, City University of New York, New York, New York, United States of America

☢ These authors contributed equally to this work
* eunjin.erica.cho@gmail.com

## Abstract

Antifungal resistance is growing increasingly more common due to the widespread use of limited number of antifungal compounds classes. Plant extracts have been used and studied for thousands of years as antifungal therapeutics alone or in combination with other natural products. This study investigated the synergistic effects of combining ethanolic extracts from nine plants with documented antifungal activity to identify natural and more powerful antifungal treatments against *Candida albicans*. Using checkerboard microdilution assays, 11 out of 15 combinations exhibited additive or synergistic interactions (fractional inhibitory concentration index, FICI < 1). The strongest synergy was observed between *Alpinia officinarum* and *Hydrastis canadensis* with $MIC_{90}$ FICI = 0.08 and $MIC_{50}$ FICI = 0.05. Combinations involving *H. canadensis*, *Eucalyptus globulus,* and *Punica granatum* produced the most synergistic effects with other tested extracts and with each other. Combining putative active compounds from each of these three extracts demonstrated synergistic antifungal activity, with the strongest synergy observed with berberine (from *H. canadensis*) and punicalagin (from *P. granatum)* with $MIC_{90}$ FICI = 0.31 and $MIC_{50}$ FICI = 0.13. Eucalyptol did not produce any significant antifungal activity so *E. globulus* extract was fractionated to identify its main antifungal compounds. UPLC-MS analysis determined that the most active fractions were primarily made up of hydrolysable tannins which produced strong synergy when combined to berberine with $MIC_{90}$ FICI = 0.31 and $MIC_{50}$ FICI = 0.25. The combinations of berberine with punicalagin and berberine with the *E. globulus* high tannin fraction F5 displayed antifungal activity against *C. albicans* with $MIC_{90}$ concentrations of 2–16 µg/mL, which are comparable to $MIC_{90}$ concentration for econazole of 0.5–8 µg/mL. These results suggest that phytochemical mixtures containing different classes of antifungal compounds can approach the efficacy of commercial antifungals and may serve as effective alternatives.

**Data availability statement:** All relevant data are within the manuscript and its Supporting Information files.

**Funding:** This work was partially supported by NIH / ODS / NCCIH Botanical Center Grant (P50 AT002776 to IR), and the NJ Agricultural Experiment Station of Rutgers, The State University of NJ. Funding support for KA and JH was provided by the National Center for Complementary and Integrative Health through training grant 5T32AT004094. There was no additional external funding received for this study. The funders had no role in study design, data collection and analysis, decision to publish, or preparation of the manuscript.

**Competing interests:** The authors have declared that no competing interests exist.

## Introduction

Traditional herbal medicines have used plant powders and extracts to treat diverse conditions for thousands of years [1]. Use of plants continues today as 179 countries report to using traditional herbal medicines and 80% of the developing world continues to at least partially rely on traditional medicinal plants for therapeutics, including for infectious diseases [2]. Much of this traditional usage of botanical medicines is based on assumed synergistic interactions within mixtures of several ingredients such as in traditional Chinese medicines where multiple plants are commonly combined [3]. Multiple studies corroborate that combinations of botanical extracts have better clinical efficacy than single compounds or extracts through synergy, the combined effect exceeding the sum of its parts. For example, Elfawal et al. [4] demonstrated that a single dose of *Artemisia annua* extract is more potent against malaria parasites than a comparable dose of pure artemisinin.

The spread of antibiotic-resistant pathogens has sustained interest in plant-based therapeutics [5–6]. Resistances to antifungals developed in many microorganisms due to expanding use of single compound/ single target antifungals. Compared to bacteria, fungi pose a unique issue because of their evolutionary eukaryotic similarity to humans and the limited number of antifungals classes used to treat them, such as allylamines, azoles, echinocandins, polyenes, and pyrimidine analogs. Overuse of these antifungals has led to rising resistances to one or multiple antifungal classes, prompting a shift to combination therapies [2]. *Candida albicans* is a pathogenic fungus that is increasingly common among fungal infections and a widespread contaminant in hospitals and medical devices [7–8]. Azoles like fluconazole, voriconazole, and econazole target ergosterol biosynthesis in fungi [9] and are commonly used against candidiasis due to limited mammalian cytotoxicity compared to other antifungals like amphotericin B [10]. However, many *Candida* species develop azole resistances by modifications in the ergosterol biosynthesis pathway, namely overexpression and mutations in *ERG11* that reduce binding between Erg11p and fluconazole [9,11,12].

Interactions between compounds within complex plant metabolite mixtures can be positive and synergistic via complimentary modes of action, enhancing potency of actives, and preventing resistance development in bacteria and fungi [13–14]. It has been hypothesized that the evolution of plants' complex chemical composition is, at least in part, driven by the need to exploit synergistic interactions among multiple phytochemicals to enhance defense against microbial pathogens and herbivores [13]. Phytochemical interactions have evolved as protective measures against pathogens and pathogen resistance and can produce broader therapeutic effects than single target compounds. Synergy between plant extracts can affect several pathogenic targets at once, further preventing the development of antibiotic resistances. Combinations of plant extracts have shown synergistic fungicidal activity against *C. albicans*, often with lower minimum inhibitory concentration (MIC) than their single components [15–17]. Research also supports the synergistic effects of plant extracts with synthetic antibiotics, such as potentiating the effects of fluconazole with berberine against fluconazole resistant *C. albicans*, by promoting uptake of berberine, resulting in DNA damage and cell cycle arrest [18–20].

The main objective of this study was to characterize the possible synergies between ethanolic extracts of nine plants previously reported to have strong antifungal activity and to identify putative actives responsible for the observed synergism. *Alpinia officinarum, Eucalyptus globulus, Humulus lupulus, Hydrastis canadensis, Matricaria chamomilla, Phellodendron amurense, Punica granatum, Scutellaria baicalensis,* and *Viola tricolor* containing antifungal bioactives were used in our study.

## Materials and methods

### Metabolomic analysis

A Bruker Daltonics maXis-II UHR-ESI-QqTOF mass spectrometer coupled to a Thermo Scientific Ultimate 3000 UHPLC system was used for analytical measurements. All samples were run in duplicates, with blank runs inserted between each sample to minimize cross-contamination. Each injection consisted of 10 µL of sample (1 mg/mL) applied to an Agilent Acclaim 120 C18 column (2.1 mm × 150 mm, 2.2 µm) and maintained at 30 °C with a flow rate of 150 µL/min.

The gradient elution began with 2% solvent B (acetonitrile with 0.15% formic acid) and 98% solvent A (water with 0.15% formic acid) for the first 2 minutes, followed by a linear gradient increase to 40% solvent B over 20 minutes, then an increase to 98% solvent B over the next 10 minutes, and a final hold at 98% solvent B for an additional 10 minutes.

Mass spectrometry data were acquired over an m/z range of 50–1300 in negative-ion mode electrospray ionization. Raw data were processed using Bruker's MetaboScape 2024b software and interpreted alongside several metabolomics databases, including the Bruker MetaboBASE Personal Library 3.0 (https://store.bruker.com/products/bruker-metabobase-personal-library-3-0), the MassBank of North America (MoNA) with LipidBlast 2022 (https://mona.fiehnlab.ucdavis.edu/), and the Human Metabolome Database (HMDB) (https://www.hmdb.ca/). Metabolites with structural identifications were classified into biosynthetic pathways, superclasses, and classes using NPClassifier [21]. Relative abundances were determined by summing ion intensities within each structural class.

### UPLC-MS/MS analysis

Samples of crude ethanolic extracts were separated and analyzed by a UPLC-MS/MS system including the Dionex® UltiMate 3000 RSLC ultra-high-pressure liquid chromatography system, with a workstation with ThermoFisher Scientific's Xcalibur v. 4.0 software package. After photodiode array detector, the eluent was guided to a Q Exactive Plus Orbitrap high-resolution high-mass-accuracy mass spectrometer (MS). Mass detection was full MS scan with low energy collision induced dissociation (CID) from 100 to 1000 m/z in positive and negative ionization mode with electrospray (ESI) interface. The mass resolution was 140,000. Substances were separated on a PhenomenexTM Luna C8 reverse phase column.

### Plant material

Dried plant materials were purchased from various vendors for assays. *Alpinia officinarum, Eucalyptus globulus, Humulus lupulus, Hydrastis canadensis, Matricaria chamomilla, Phellodendron amurense, Punica granatum, Scutellaria baicalensis,* and *Viola tricolor* were selected for the study due to their various and well-documented antifungal bioactive compounds (Table 1). *A. officinarum* rhizomes were purchased from Ecowise (Ahmedabad, Gujarat, India), *E. globulus* leaves*, H. canadensis* rhizomes*, S. baicalensis* rhizome*,* and *M. chamomilla* flowers were purchased from Starwest Botanicals (Sacramento, California, USA), *H. lupulus* flowers and *V. tricolor* whole plants were purchased from Biokoma (Lake Villa, Illinois, USA), and *P. granatum* pericarp powder and *P. amurense* bark were purchased from Naturevibe Botanicals (Rahway, New Jersey, USA). Plant materials were stored in −20 °C and ground to a fine powder before extraction. Botanical verification of *P. granatum* powder was tested by assessing bioactivity of whole fruit pericarp. Pericarp was freeze dried and made to extract following the same procedure as the powder.

**Table 1. Summary of plants used in this study and their putative antifungal actives.**

| Species Name | Common Name | Putative Actives | Part |
| --- | --- | --- | --- |
| *Alpinia officinarum* | Galangal | Galangin, ceftazidime, quercetin [22–24] | Rhizome |
| *Eucalyptus globulus* | Blue gum | Eucalyptol, limonene, α-pinene, α-terpineol [25–26] | Leaf |
| *Humulus lupulus* | Hops | α- and β-bitter acids, terpenes, sesquiterpenes [27–29] | Cone |
| *Hydrastis canadensis* | Goldenseal | Berberine alkaloids [20,30,31] | Rhizome |
| *Matricaria chamomilla* | Chamomile | α-bisabolol, α-pinene, limonene [32–33] | Flower |
| *Phellodendron amurense* | Amur cork tree | Palmatine, berberine [34–35] | Bark |
| *Punica granatum* | Pomegranate | Ellagitannins, proanthocyanidins [36–38] | Pericarp |
| *Scutellaria baicalensis* | Chinese skullcap | Baicalin, wogonin flavonoids [39–41] | Rhizome |
| *Viola tricolor* | Wild pansy | Cyclic peptides (cycloids) [42] | Whole |

## Reagents, chemicals and fungal strain

The *C. albicans* used was strain 10231 purchased from the American Type Culture Collection (ATCC) (Manassas, VA, USA). *C. albicans* strain 10231 was chosen because it is widely studied and characterized as a strain for QC standards for various assays by CLSI and EUCAST, whose methods were used to base our own study methodology on. They were grown on plates of Potato Dextrose Agar (P6685) supplemented with streptomycin (S9137) purchased from Sigma-Aldrich (St. Louis, MO, USA). For the in vitro antifungal assays, Roswell Park Memorial Institute (RPMI) 1640 media powder (R6504) from Sigma-Aldrich (St. Louis, MO, USA) was used with 96-well plates (FB012931) from Thermo Fisher Scientific (Waltham, MA, USA). Fungal inoculum was prepared with 0.5 MacFarland Standard (89426−218) from VWR (Radnor, PA, USA). Positive control used in assays was econazole purchased from Sigma-Aldrich (E4632). Botanical extracts were fractionated using Sephadex LH-20 (LH20100) from Sigma-Aldrich.

## Extraction

All extracts were prepared in accordance with the rapid extraction method from Skubel et al. (2018) [43] with modifications. In summary, all dried plant material was ground in Dremel rotary tool or coffee grinders into a fine powder. A 200 mg portion of the powder was shaken in 5 mL of 70% ethanol at 70 rpm for 10 min and filtered through a sieve before being centrifuged at 2000 rpm for 10 min. Supernatant was transferred to a pre-weighed glass vial and dried in a SpeedVac vacuum concentrator for 24 hrs. Vials were weighed after drying and resuspended in 70% ethanol to bring to a starting concentration of 20 mg/mL.

## Extract fractionation

Extracts were separated in a Sephadex LH-20 column with protocol adapted from Bellesia et al. (2014) [44]. To prepare columns with a bed volume of 125 mL, two volumes of 50% v/v acetone followed by two volumes of 80% v/v ethanol were used as a wash, with additional conditioning with one bed volume of 80% v/v ethanol immediately prior to a run. Extracts were added to the column and eluted with two volumes of 80% ethanol to elute anthocyanins and other smaller molecule phenolic compounds before eluting the bound fraction including larger phenolic tannins with two volumes of 50% acetone [44–45]. The eluents were pooled into five fractions consisting of three 80% ethanol fractions and two 50% acetone fractions. These fractions were dried and brought to a working concentration of 1 mg/mL for antifungal testing and LC-MS analysis.

## Thin layer chromatography (TLC) agar overlay bioautography

Fractions were made to 1 mg/mL and extracts were 2-fold diluted 5 times from 10 mg/mL to 0.31 mg/mL. Each fraction and extract dilution were spotted onto a TLC plate and an agar overlay bioautography method adapted from Rahalison

et al. (1991) [46] was used. *C. albicans* was inoculated into melted potato dextrose agar and 5 mL was pipetted onto the spotted TLC plates, ensuring all spots were covered and no agar ran off the edge of the TLC plates. Plates with agar were left to solidify for 10 minutes before incubating at 37 °C for 24 hrs. TLC plates were then dipped in a 5 mL pool of 1 mg/mL 3-(4,5-dimethylthiazol-2-yl)-2,5-diphenyltetrazolium bromide (MTT) and left to incubate again at 37 °C for 24 hrs, letting the MTT color to develop. The appearance of purple color indicates the presence of the metabolically active cells that reduce the yellow tetrazolium salt to a purple product.

### In-vitro antifungal assays

The microdilution assay was performed in accordance with methods from the European Committee on Antimicrobial Susceptibility (EUCAST) with modifications [47]. A suspension of *C. albicans* in sterilized DI water equivalent to the concentration of 0.5 MacFarland Standard, or $1-5 \times 10^6$ CFU/mL was diluted by ten to a working concentration of $1-5 \times 10^5$ CFU/mL for all antifungal assays and tested in a checkerboard assay as shown in Fig 1. To determine starting concentrations for checkerboard synergy assays, the MIC was first determined for each botanical extract. Extracts were 2-fold serially diluted eight times from 20 mg/mL to 0.16 mg/mL, with 100 μL of each dilution applied to a 96-well plate in triplicate. The 96-well plates were dried overnight so ethanol could evaporate from the extracts, leaving only a thin layer of extracted botanical material. Once the ethanol had fully evaporated, dried extracts were reconstituted in 100 μL of RPMI 1640 media and inoculated with 100 μL of *C. albicans* cell suspension, making a working concentration of 10 mg/mL to 0.078 mg/mL. Absorbances was read at 530 nm using a BioTek Synergy HT Multi-Detection Microplate Reader before and after a 24 hr incubation at 37 °C. Inhibition percentages were calculated with:

$$\text{Inhibition } \% = \left( 1 - \left( \frac{OD_{24H} - OD_{0H}}{ODC_{24H} - ODC_{0H}} \right) \right) * 100$$

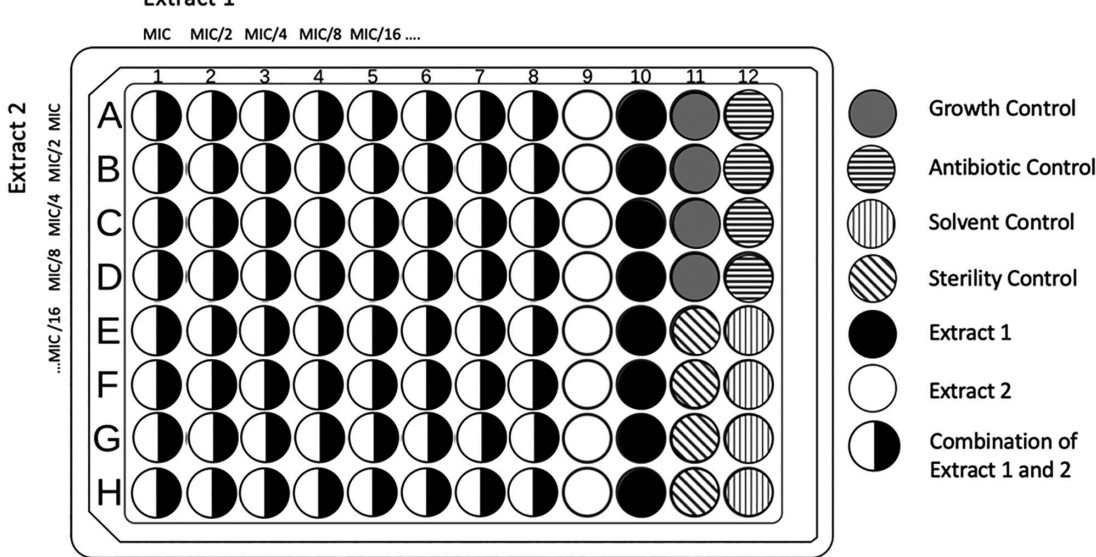

**Fig 1. Checkerboard assay layout on a 96-well plate.** Two extracts were combined in concentrations starting at the calculated $MIC_{90}$ and in subsequent 2-fold diluted concentrations. Antifungal used as control was econazole at 30 μg/mL. Solvent control used 70% ethanol. Growth control was *C. albicans* inoculum with RPMI 1640 media with no treatment. Columns 9 and 10 contained only one extract at concentrations starting at the calculated $MIC_{90}$ and 2-fold serially diluted.

MICs were determined as the lowest concentration that reached at least 90% or 50% inhibition.

Checkerboard assays were used to find synergies between two extracts, as shown in Fig 1. Extract concentrations were determined by $MIC_{90}$, with the highest concentration being the $MIC_{90}$ and each subsequent concentration halved for eight concentrations. Each well received 100 µL of each concentration from both botanical extracts and dried overnight, or until the ethanol was fully evaporated, up to 24 hrs. Assays were inoculated as previous MIC testing. Synergy was calculated using the fractional inhibitory concentration index (FICI) for each extract combination [48]. The FICI data was interpreted using the following criteria: synergistic: FICI < 0.5, additive: 0.5 ≤ FICI < 1, neutral: 1 ≤ FICI < 2, and antagonistic: FICI ≥ 2.

$$FICI = \left[\frac{MIC_{A(with\ B)}}{MIC_{A(alone)}}\right] + \left[\frac{MIC_{B(with\ A)}}{MIC_{B(alone)}}\right]$$

All assays were performed in a 96-well plate with growth control containing 100 µL RPMI 1640 and 100 µL inoculum, sterility control containing 200 µL RPMI 1640, solvent control containing 100 µL of 70% ethanol dried overnight, 100 µL RPMI 1640 and 100 µL inoculum, and a positive control of 190 µL RPMI 1640, inoculum and 10 µL of 30 µg/mL econazole, as shown in Fig 1. All assays were run in triplicate and MICs and FICIs were calculated as an average of the three treatments and controls.

### Statistical analysis

All statistical analyses were performed on GraphPad PRISM. Data were presented as the mean of triplicates ± standard deviation of the mean. Data for S1 Fig were evaluated using analysis of variance (ANOVA) and Tukey's HSD post-hoc test for multiple comparisons to display significant differences in mean inhibition percentage for each combination treatment. Each treatment was labelled with a difference letter based on significance between treatments, with different letters representing significant differences, with $\alpha = 0.05$.

### Results

Antifungal activity of ethanolic plant extracts from Table 1 were evaluated using $MIC_{90}$ and $MIC_{50}$ against *C. albicans* with microdilution in 2-fold descending concentrations starting from 10 mg/mL (Table 2). *M. chamomilla* and *V. tricolor* showed low antifungal activity in microdilution, with a $MIC_{90}$ and $MIC_{50}$ greater than 20 mg/mL and were excluded for the remainder of the study. In comparison, other extracts displayed significant activity against *C. albicans*. *A. officinarum, H. canadensis,* and *S. baicalensis* had the lowest $MIC_{90}$ at 5 mg/mL, 2.5 mg/mL, and 0.63 mg/mL respectively. Low $MIC_{50}$ included *E.*

**Table 2. $MIC_{90}$ and $MIC_{50}$ for each plant extract against *C. albicans* in broth microdilution measured with $OD_{530}$.**

| Species Name | $MIC_{90}$ (mg/mL) | $MIC_{50}$ (mg/mL) |
|---|---|---|
| *Alpinia officinarum* | 5 | 5 |
| *Eucalyptus globulus* | 10 | 1.25 |
| *Humulus lupulus* | ≥20 | 0.625 |
| *Hydrastis canadensis* | 2.5 | 2.5 |
| *Matricaria chamomilla* | ≥20 | ≥20 |
| *Phellodendron amurense* | ≥20 | 2.5 |
| *Punica granatum* | ≥20 | 5 |
| *Scutellaria baicalensis* | 0.63 | 0.31 |
| *Viola tricolor* | ≥20 | ≥20 |

*globulus, H. lupulus,* and *S. baicalensis* at 1.25 mg/mL, 0.625 mg/mL, and 0.31 mg/mL respectively. Extracts with the highest activity against *C. albicans* were reportedly flavonoid-rich such as baicalin and wogonin in *S. baicalensis* and galangin and quercetin in *A. officinarum*, or terpene-rich such as eucalyptol in *E. globulus* and various sesquiterpenes in *H. lupulus* (Table 1 and S1 File).

$MIC_{90}$ and $MIC_{50}$ values were evaluated for each extract combination using a checkerboard assay described in Britton et al. (2018) [20] to determine synergy based on the FICI as seen in Table 3 and S2 File. FICI values ranged from 0.039, highly synergistic, to 4.0, highly antagonistic. Several combinations demonstrated potent synergistic antifungal activity against *C. albicans*. The combination containing *A. officinarum* and *H. canadensis* yielded the strongest synergy with $MIC_{50}$ and $MIC_{90}$ FICI values of 0.05 and 0.08 respectively. *P. granatum* with *S. baicalensis* was strongly antagonistic with

**Table 3. Calculated FIC Index scores of plant extract combinations against *C. albicans* according to $MIC_{90}$ and $MIC_{50}$.**

| Plant Combination | Calculate with | Individual MIC (mg/mL)[a] | Combination MIC (mg/mL)[a, b] | FIC Index[c] | Outcome |
|---|---|---|---|---|---|
| *A. officinarum*/*E. globulus* | $MIC_{90}$ | 5.0/10.0 | 5.0/1.25 | 1.13 | Neutral |
| | $MIC_{50}$ | 5.0/1.25 | 0.16/1.25 | 1.03 | Neutral |
| *A. officinarum*/*H. canadensis* | $MIC_{90}$ | 5.0/2.5 | 0.078/0.16 | 0.08 | Synergistic |
| | $MIC_{50}$ | 5.0/2.5 | 0.078/0.078 | 0.05 | Synergistic |
| *A. officinarum*/*P. amurense* | $MIC_{90}$ | 5.0/20 | 2.5/20.0 | 1.50 | Neutral |
| | $MIC_{50}$ | 5.0/2.5 | 2.5/1.25 | 1.00 | Neutral |
| *A. officinarum*/*P. granatum* | $MIC_{90}$ | 5.0/20.0 | 0.63/5.0 | 0.38 | Synergistic |
| | $MIC_{50}$ | 5.0/5.0 | 0.16/0.16 | 0.063 | Synergistic |
| *A. officinarum*/*S. baicalensis* | $MIC_{90}$ | 5.0/0.63 | 5.0/0.078 | 1.13 | Neutral |
| | $MIC_{50}$ | 5.0/0.31 | 2.5/0.020 | 0.56 | Additive |
| *E. globulus*/*H. canadensis* | $MIC_{90}$ | 10.0/2.5 | 0.078/0.31 | 0.13 | Synergistic |
| | $MIC_{50}$ | 1.25/2.5 | 0.039/0.020 | 0.039 | Synergistic |
| *E. globulus*/*P. amurense* | $MIC_{90}$ | 10.0/20.0 | 1.25/1.25 | 0.19 | Synergistic |
| | $MIC_{50}$ | 1.25/2.5 | 0.31/0.31 | 0.38 | Synergistic |
| *E. globulus*/*P. granatum* | $MIC_{90}$ | 10.0/20.0 | 5.0/10.0 | 1.0 | Neutral |
| | $MIC_{50}$ | 1.25/5.0 | 0.31/1.25 | 0.5 | Additive |
| *E. globulus*/*S. baicalensis* | $MIC_{90}$ | 10.0/0.63 | 1.25/0.31 | 0.63 | Additive |
| | $MIC_{50}$ | 1.25/0.31 | 0.31/0.020 | 0.31 | Synergistic |
| *H. canadensis*/*P. amurense* | $MIC_{90}$ | 2.5/20.0 | 1.25/0.16 | 0.51 | Additive |
| | $MIC_{50}$ | 2.5/2.5 | 0.63/0.63 | 0.50 | Additive |
| *H. canadensis*/*P. granatum* | $MIC_{90}$ | 2.5/20.0 | 1.25/0.040 | 0.50 | Additive |
| | $MIC_{50}$ | 2.5/5.0 | 0.16/0.040 | 0.070 | Synergistic |
| *H. canadensis*/*S. baicalensis* | $MIC_{90}$ | 2.5/0.63 | 1.25/0.020 | 0.52 | Additive |
| | $MIC_{50}$ | 2.5/0.31 | 0.16/0.16 | 0.56 | Additive |
| *P. amurense*/*P. granatum* | $MIC_{90}$ | 20.0/20.0 | 0.63/1.25 | 0.094 | Synergistic |
| | $MIC_{50}$ | 2.5/5.0 | 0.16/0.16 | 0.094 | Synergistic |
| *P. amurense*/*S. baicalensis* | $MIC_{90}$ | 20.0/0.63 | 20.0/0.63 | 2.0 | Antagonistic |
| | $MIC_{50}$ | 2.5/0.31 | 0.63/0.31 | 2.13 | Antagonistic |
| *P. granatum*/*S. baicalensis* | $MIC_{90}$ | 20.0/0.63 | 0.040/2.5 | 4.0 | Antagonistic |
| | $MIC_{50}$ | 5.0/0.31 | 0.040/1.25 | 4.0 | Antagonistic |

[a]The $MIC_{90}$ and $MIC_{50}$ are displayed as Species 1/Species 2 for both individual and combination MIC.

[b]Selected Combination MIC values were the combination that produced the best FICI.

[c]Synergistic = FICI < 0.5, additive = 0.5 ≤ FICI < 1, neutral = 1 ≤ FICI < 2, antagonistic = FICI ≥ 2.

both $MIC_{90}$ and $MIC_{50}$ FICI values of 4.0. Some extract combinations displayed antifungal inhibition on par with commercial antifungal econazole at a concentration of 30 µg/mL (S1 Fig).

Of the six plant extracts, *H. canadensis*, *P. granatum*, and *E. globulus* had the most combinations that resulted in synergistic or additive FICI values and least neutral and antagonistic combinations based on both $MIC_{90}$ and $MIC_{50}$. With $MIC_{90}$, *H. canadensis* had 2 synergistic and 3 additive combinations, *P. granatum* had 2 synergistic and 1 additive combination, and *E. globulus* also had 2 synergistic and 1 additive combinations. With $MIC_{50}$, *H. canadensis* had 3 synergistic and 2 additive combinations, *P. granatum* had 3 synergistic and 1 additive combinations, and *E. globulus* also had 3 synergistic and 1 additive combinations (Fig 2). The only antagonistic combinations were those containing *S. baicalensis*, with FICI values ranging from 2.0 to 4.0 with *P. amurense* and *P. granatum* respectively (Table 3 and Fig 2).

To confirm the identities of the putative anti-*C. albicans* actives from the most active extracts and to assess the synergies between them, berberine for *H. canadensis*, punicalagin for *P. granatum,* and eucalyptol for *E. globulus* were tested. Berberine and punicalagin displayed a stronger growth inhibition of *C. albicans* than corresponding concentrations of crude extract, with $MIC_{90}$ values of 0.06 mg/mL for berberine and of 0.03 mg/mL for punicalagin compared to $MIC_{90}$ values of 2.5 mg/mL for *H. canadensis* extract and ≥ 20 mg/mL for *P. granatum* extract (Fig 3 and Table 2). These data confirm that these compounds are likely responsible for a significant part of antifungal activity of the corresponding extracts. Punicalagin in *P. g*ranatum pericarp crude extract was quantified using LC-MS and found to be 26.8 mg/g of extract dry weight, which is consistent with the literature reports [49]. *H. c*anadensi*s* has been widely studied, with commercial *H. canadensis* root products reported to contain about 0.82% to 4% berberine w/w, or 8–58 mg/g [50–51]. Although eucalyptol displayed stronger antifungal activity than the crude extract, with a $MIC_{90}$ of 1 mg/mL compared to 10 mg/mL, LC-MS analysis was unable to detect any eucalyptol, eliminating it as a representative antifungal compound in our *E. globulus* extract (S1 Table, S2 Fig). Eucalyptol, a volatile compound, is most abundant in *E. globulus* essential oils comprising 45% − 85% [52] whereas ethanolic extracts contain less than 18% eucalyptol [53]. Since our extraction procedure includes drying to evaporate ethanol this likely removed most if not all volatile compounds.

To identify other potential actives responsible for *E. globulus* crude extract antifungal activity, the extract was fractionated into 25 fractions using a Sephadex LH-20 column gravity chromatography. Fraction bioactivity was tested against *C. albicans* using TLC agar overlay bioautography (S3 Fig). The observed zones of inhibition of MTT-stained *C. albicans*

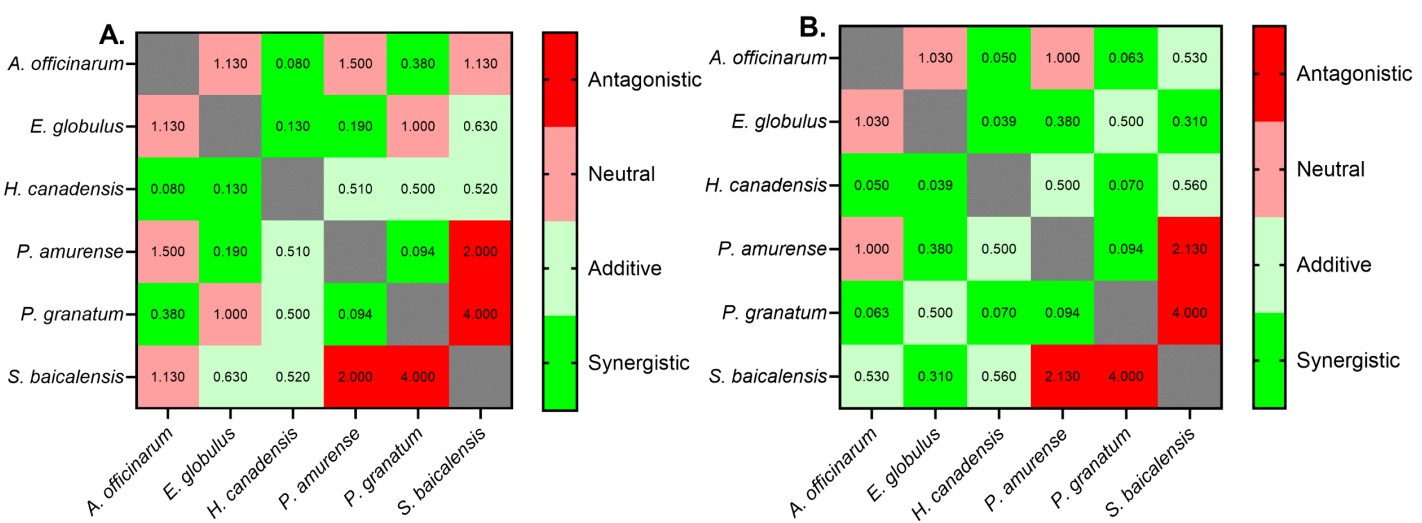

**Fig 2. Heatmap matrices of FICI values for each plant extract combination. (A)** $MIC_{90}$ values and **(B)** $MIC_{50}$ values. Green indicates strongly synergistic and red indicates strongly antagonistic. Synergistic = FICI < 0.5, additive = 0.5 ≤ FICI < 1, neutral = 1 ≤ FICI < 2, antagonistic = FICI ≥ 2.

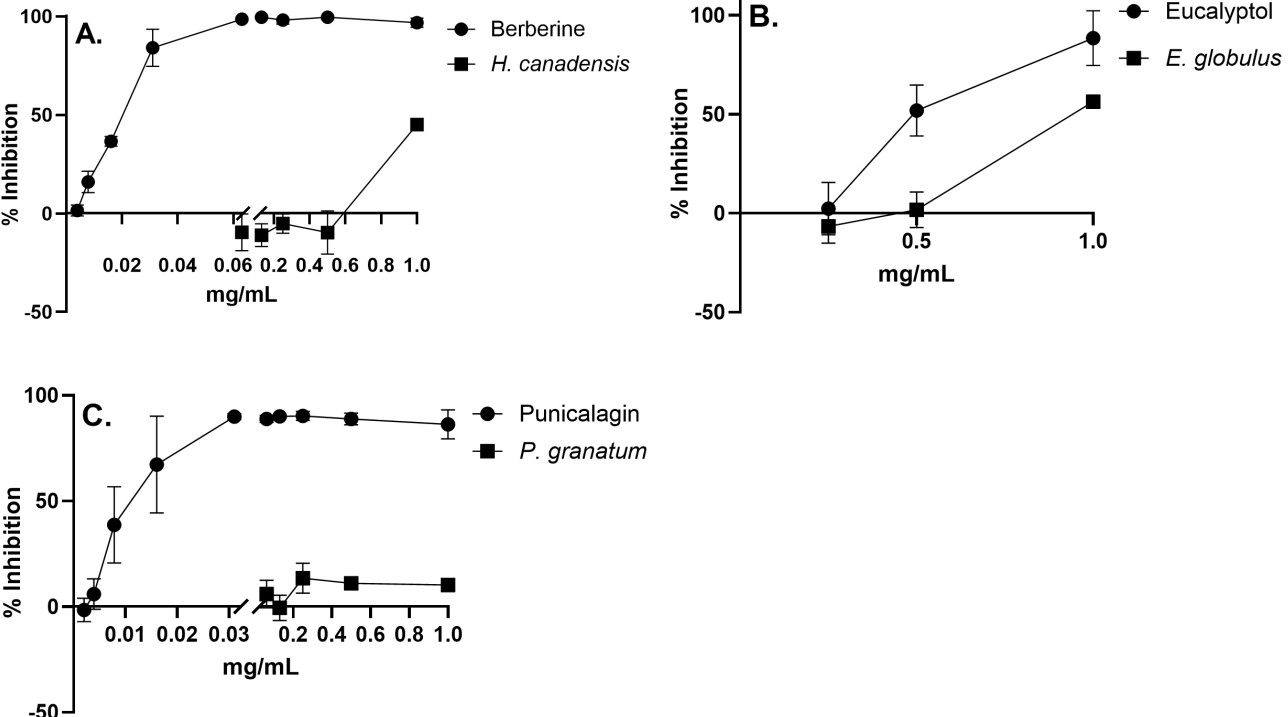

**Fig 3. Comparison of antifungal activity as % inhibition of *C. albicans* growth between plant extracts and their putative active compounds.** Antifungal activity of **(A)** berberine and *H. canadensis,* **(B)** eucalyptol and *E. globulus,* and **(C)** punicalagin and *P. granatum.* For all data points n = 3. Bars represent ±S.D.

indicated that acetone fractions A2-A11 produced the strongest bioactivity. All fractions were combined in pools of five F1-F5, with the strongest bioactivity in fractions F4 (A2 – A6) and F5 (A7 – A11). F5 displayed stronger bioactivity against *C. albicans* compared to F4 (S3 Fig).

LC-MS/MS analysis was used to putatively identify and quantify metabolites in the *E. globulus* crude extract and its fractions F4 and F5. All three were mainly composed of shikimates and phenylpropanoids, along with smaller amounts of polyketides, terpenoids, fatty acids and carbohydrates (S4A Fig). Of the shikimates and phenylpropanoids in the crude extract, a large majority were flavonoids and phenolic acids, with some additional phenylpropanoids, coumarins, phenylethanoids and lignins (S4B Fig). The crude extract contained a mixture of flavonoids, phenolic acids, and hydrolysable tannins, with kaempferol 3-glucuronoside and quercetin 3-O-glucuronide being the most abundant, present at levels at least twice as high as any other metabolite (S1 Table, S2 Fig). Fractions F4 and F5 displayed similar profiles to the crude extract but were highly enriched in phenolic acids, particularly hydrolysable tannins and their derivatives, which accounted for approximately 80% and 95% of their composition, respectively. The most abundant putative compounds in F4 and F5 included hydrolysable tannins camelliin A and 1,2,3,6-tetragalloylglucose, with other highly abundant putative ellagitannins and gallotannins. Notably, camelliin A and 1,2,3,6-tetragalloylglucose were the most abundant shared compounds across the crude extract, F4, and F5 (S1 Table, S2 Fig).

The putative actives of the most active antifungal extract used in this study were combined to assess synergy against *C. albicans*. Berberine combined with punicalagin produced synergistic anti-*C. albicans* effects with $MIC_{90}$ FICI = 0.31 and $MIC_{50}$ FICI = 0.13, but berberine and punicalagin combined with eucalyptol displayed no synergisms, further indicating that eucalyptol was not the source of synergy within the crude extract combinations (Table 4). However, the tannin-rich

F5 demonstrated stronger synergisms than the eucalyptol. Combining F5 and berberine showed synergy with $MIC_{90}$ FICI = 0.31 and $MIC_{50}$ FICI = 0.25. F5 with punicalagin showed additive activity with FICI of 0.64 and 0.63 with $MIC_{90}$ and $MIC_{50}$ respectively (Table 4). The synergistic combinations displayed strong antifungal potency – the berberine and punicalagin combination had combinatorial $MIC_{90}$ concentrations of 0.016 mg/mL and 0.0020 mg/mL (Table 4).

## Discussion

The goal of this study was to assess the synergies between ethanolic extracts of highly active antifungal botanicals and to relate these synergies to their active compounds. We observed that more extract combinations demonstrated synergistic or additive effects when assessed using $MIC_{50}$ rather than $MIC_{90}$ values. $MIC_{50}$ values were achieved by most combinations of the selected crude plant extracts at the concentrations used. This is likely because plants, while co-evolving with microbial pathogens, have relied on synergy from complex mixtures of bioactive compounds, rather than on a single powerful antibiotic [14,54]. While the majority of plant extracts in this study already exhibit strong antifungal activity against *C. albicans*, our data suggest that combining these extracts may result in synergistic antifungal activity that is much greater than any individual extract alone, particularly at the $MIC_{50}$ level.

Synergies between plant extracts usually arise from the combinations of different compound classes [54]. Berberine, the antifungal component of *H. canadensis*, may intercalate into fungal cell walls and membranes, disrupting their integrity [55], and inhibiting ergosterol biosynthesis by targeting lanosterol 14α-demethylase (CYP51) [56]. Berberine has also been documented to modify functions of fungal efflux pumps Mdr1p and Cdr2p, causing berberine accumulation in the cells resulting in disruption of mitochondria [19,57,58]. Additionally, berberine can cause DNA damage and cell cycle arrest, affect DNA replication, transcription and maintenance [19]. Interestingly, all combinations with *H. canadensis* we tested displayed either additive or synergistic antifungal activity with FICI values of 0.52 to 0.0039 (Table 3 and Fig 2).

We observed synergy between berberine and hydrolysable tannins such as punicalagin from *P. granatum* and possibly other hydrolysable tannins (F5 from *E. globulus*) (Table 4). Hydrolysable tannins, including gallotannins and ellagitannins like punicalagin, exhibit antifungal activity against fungi such as *C. albicans* [37,59]. Their antimicrobial effects are largely attributed to metal chelation, reducing bioavailable $Fe^{2+}$ and $Fe^{3+}$, and irreversible protein-binding interactions that disrupt fungal adhesins, cell wall polypeptides, and membrane proteins [37,60,61]. These actions destabilize membranes and cell walls [62–63] and can deactivate fungal digestive enzymes [64–65]. Differences in MIC values among tannins likely stem

**Table 4. Calculated FIC Index scores of pure compound or fraction combinations of select extracts against *C. albicans* according to $MIC_{50}$ and $MIC_{90}$ in mg/mL.**

| Compound Combination | Calculated with | Individual MIC (mg/mL) | Combination MIC (mg/mL)[a] | FIC Index[b] | Outcome |
|---|---|---|---|---|---|
| Berberine/Eucalyptol | $MIC_{90}$ | 0.063/1.0 | 0.063/0.50 | 1.50 | Neutral |
| | $MIC_{50}$ | 0.031/0.50 | 0.031/0.13 | 1.25 | Neutral |
| Berberine/Punicalagin | $MIC_{90}$ | 0.063/0.031 | 0.016/0.0020 | 0.31 | Synergistic |
| | $MIC_{50}$ | 0.031/0.016 | 0.0020/0.0039 | 0.13 | Synergistic |
| Eucalyptol/Punicalagin | $MIC_{90}$ | 1.0/0.031 | 0.50/0.016 | 1.02 | Neutral |
| | $MIC_{50}$ | 0.50/0.016 | 0.50/0.008 | 1.50 | Neutral |
| Berberine/F5 | $MIC_{90}$ | 0.063/0.13 | 0.016/0.016 | 0.31 | Synergistic |
| | $MIC_{50}$ | 0.031/0.063 | 0.0039/0.008 | 0.25 | Synergistic |
| Punicalagin/F5 | $MIC_{90}$ | 0.031/0.13 | 0.016/0.016 | 0.64 | Additive |
| | $MIC_{50}$ | 0.016/0.063 | 0.008/0.008 | 0.63 | Additive |

[a]Selected Combination MIC values were the combination that produced the best FIC

[b]Synergistic = FICI < 0.5, additive = 0.5 ≤ FICI < 1, neutral = 1 ≤ FICI < 2, antagonistic = FICI ≥ 2.

from variations in galloylation and size, which influence their ability to chelate metals and bind proteins [60,66]. Highly galloylated tannins tend to show stronger protein inhibition, such as against SrtA and topoisomerases, although larger tannins may have reduced cell penetration [59]. Punicalagin, in particular, inhibits yeast topoisomerases with an $IC_{50}$ of 14.7 µM, greater than that of camptothecin which has an $IC_{50}$ of 17.8 µM [59].

Berberine and some hydrolysable tannins likely act synergistically due to their complementary mechanisms of action on cell wall, membranes, and mitochondrial metabolism. Our data confirms that berberine combined with hydrolysable tannin-rich F5 or punicalagin form potent antifungal mixtures comparable to antifungal drugs. The combination of punicalagin and F5 resulted in an additive FICI score likely because the majority of compounds in F5 were also hydrolysable tannins. $MIC_{90}$ values against *C. albicans* for commercial synthetic antifungals such as econazole against *C. albicans* ranged from 0.5 µg/mL to 8 µg/mL [67]. While most combinations usually did not reach the potency of econazole, some such as berberine with punicalagin and berberine with F5, were respectively able to reach a similar antifungal efficacy of around 16 and 2 µg/mL as well as 16 and 16 µg/mL for $MIC_{90}$ values (Table 4).

The growing resistance of *C. albicans* to conventional antifungals underscores the need for alternative therapies. The observed synergy between berberine and hydrolysable tannins such as punicalagin or F5 highlights the potential of plant-derived combinations to inform the development of antifungal therapeutics of the future. These findings suggest that natural compound mixtures may supplement or even replace the existing treatments. Phytochemical combinations may also offer advantages in accessibility, especially in resource-limited settings, and could reduce the risk of resistance development due to their multi-targeted effects. Overall, these results support further exploration of synergistic plant extract formulations as viable antifungal therapies.

## Conclusions

Among all tested extracts and their combinations, the mixture of *H. canadensis* extract with extracts from *P. granatum* or *E. globulus* exhibited the most potent and synergistic antifungal activity, approaching that of standard antifungal drugs such as econazole. Similar synergies were observed when the actives from these extracts, berberine and hydrolysable tannins, were combined. These findings highlight the potential of using multi-component plant-based therapeutics to target *C. albicans* through complementary mechanisms of action, including membrane disruption, metal chelation, intracellular enzyme inhibition, and mitochondria damage.

## Supporting information

**S1 Fig. Activity of extracts alone and in combination against *C. albicans.*** Extract combinations include (A) *A. officinarum* and *E. globulus* (B) *A. officinarum* and *H. canadensis* (C) *A. officinarum* and *P. amurense* (D) *A. officinarum* and *P. granatum* (E) *A. officinarum* and *S. baicalensis* (F) *E. globulus* and *H. canadensis* (G) *E. globulus* and *P. amurense* (H) *E. globulus* and *P. granatum* (I) *E. globulus* and *S. baicalensis* (J) *H. canadensis* and *P. amurense* (K) *H. canadensis* and *P. granatum* (L) *H. canadensis* and *S. baicalensis* (M) *P. amurense* and *P. granatum* (N) *P. amurense* and *S. baicalensis* (O) *P. granatum* and *S. baicalensis.* Treatment concentrations were selected based on best FICI scores for both $MIC_{90}$ and $MIC_{50}$ and econazole concentration used was 30 µg/mL. Each bar is labelled with a letter indicating significance. For all combinations n = 4 and significance was determined using p-value 0.05. Error bars represent ±S.D. Abbreviations: GC – Growth Control, *A.o – A. officinarum, E.g – E. globulus, H.c – H. canadensis, P.a – P. amurense, P.g – P. granatum, S.b – S. baicalensis.*
(TIF)

**S2 Fig. Relative abundance of top 10 most abundant metabolites for *E. globulus* extract, F4 and F5.** *Indicates truncated metabolite names. (2S,4S,5R,6S)…oxane-2-carboxylic acid represents: (2S,4S,5R,6S)-6-[3,4-dihydroxy-5-(3,4,5-trihydroxybenzoyl)oxybenzoyl]oxy-4,5-bis[(3,4,5-trihydroxybenzoyl)oxy]oxane-2-carboxylic acid.

[(2R,3S,4S,5R,6S)…oxyoxan-3-yl] 3,4,5-trihydroxybenzoate represents: [(2R,3S,4S,5R,6S)-4,5-bis[[3,4-dihydroxy-5-(3,4,5-trihydroxybenzoyl)oxybenzoyl]oxy]-2-[[3,4-dihydroxy-5-(3,4,5-trihydroxybenzoyl)oxybenzoyl]oxymethyl]-6-(3,4,5-trihydroxybenzoyl)oxyoxan-3-yl] 3,4,5-trihydroxybenzoate.
(TIF)

**S3 Fig. Agar overlay bioautography of *E. globulus* crude extracts and fractions.** Top – TLC plates with spots of extract or fractions. Bottom – agar overlay stained with MTT, with purple indicating presence of *C. albicans*. Negative control is a spot of 70% ethanol. All values for the crude extract spots are in mg/mL.
(TIF)

**S4 Fig. Distribution of metabolites in *E. globulus* crude extract (Euc_1), and two fractions of the crude extract (Euc F4 and Euc F5).** (A) Composition of natural product biosynthetic pathways, and (B) shikimate and phenylpropanoid superclasses from A.
(TIF)

**S1 Table. Feature table including the top 10 most abundant features for *E. globulus* extract, F4, and F5.**
(XLSX)

**S1 File. Raw data and calculations for extract $MIC_{50}$ and $MIC_{90}$ values.**
(XLSX)

**S2 File. Raw data and calculations for extract $MIC_{50}$ and $MIC_{90}$ values with synergy checkerboard assays.**
(XLSX)

**S3 File. Raw data and calculations for pure compounds and fractions $MIC_{50}$ and $MIC_{90}$ values individually and with checkerboard assays.**
(XLSX)

## Acknowledgments

The authors would like to thank Dr. Joan Bennett for providing laboratory space and technical support throughout this study. We would also like to thank Jennifer Schug and Sruthi Yuvaraj for assisting with antifungal assays.

## Author contributions

**Conceptualization:** Ilya Raskin.

**Data curation:** Eunjin Cho, Joshua Henkin.

**Formal analysis:** Eunjin Cho.

**Funding acquisition:** Ilya Raskin.

**Investigation:** Eunjin Cho, Joshua Henkin, Rinat Abzalimov.

**Methodology:** Eunjin Cho, Rinat Abzalimov.

**Project administration:** Eunjin Cho, Ilya Raskin.

**Resources:** Ilya Raskin.

**Software:** Kenneth Acosta, Joshua Henkin, Rinat Abzalimov.

**Supervision:** Ilya Raskin.

**Validation:** Eunjin Cho.

**Visualization:** Eunjin Cho, Kenneth Acosta.

**Writing – original draft:** Eunjin Cho.

**Writing – review & editing:** Eunjin Cho, Kenneth Acosta, Ilya Raskin.

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
