## [Decision Letter · Decision Letter 0]

29 Aug 2025

*Candida albicans*

Dear Dr. Cho,

Thank you for submitting your manuscript to PLOS ONE. After careful consideration, we feel that it has merit but does not fully meet PLOS ONE’s publication criteria as it currently stands. Therefore, we invite you to submit a revised version of the manuscript that addresses the points raised during the review process.

We look forward to receiving your revised manuscript.

Kind regards,

Theerapong Krajaejun, M.D.

Academic Editor

PLOS ONE

Journal Requirements:

“This work was partially supported by NIH / ODS / NCCIH Botanical Center Grant (P50 AT002776 to IR), and the NJ Agricultural Experiment Station of Rutgers, The State University of NJ. Funding support for KA and JH was provided by the National Center for Complementary and Integrative Health through training grant 5T32AT004094.”

“This work was partially supported by NIH / ODS / NCCIH Botanical Center Grant (P50 AT002776 to IR), and the NJ Agricultural Experiment Station of Rutgers, The State University of NJ. Funding support for KA and JH was provided by the National Center for Complementary and Integrative Health through training grant 5T32AT004094.”

Additional Editor Comments (if provided):

Two experts in the field have reviewed your manuscript. They provided valuable feedback and suggestions for improvement, which I have reviewed and agree with. Please revise your manuscript in accordance with their comments and submit the revised version by the specified deadline to facilitate the continued evaluation of your work.

Reviewers' comments:

Reviewer's Responses to Questions

**Comments to the Author**

1. Is the manuscript technically sound, and do the data support the conclusions?

Reviewer #1: Yes

Reviewer #2: Yes

2. Has the statistical analysis been performed appropriately and rigorously?

Reviewer #1: Yes

Reviewer #2: Yes

3. Have the authors made all data underlying the findings in their manuscript fully available?

Reviewer #1: Yes

Reviewer #2: Yes

4. Is the manuscript presented in an intelligible fashion and written in standard English?

Reviewer #1: Yes

Reviewer #2: Yes

Reviewer #1: The manuscript "Synergistic antifungal effects of botanical extracts against Candida albicans" is a very interesting manuscript which explained quite detail about the interactions of many active compounds. There are some comments below.

Line 32: P. granatum should be written completely Punica granatum since it is stated for the first time in the manuscript.

Line 60: Please change word “antibiotic” to “antifungals”.

Line 88: Please verify the species name Matricularia chamomilla or Matricaria chamomilla (in Table 1).

Table 1: In my opinion, it is better to put Table 1 in Materials and Methods rather than in Introduction.

Line 97: Please write “10 µL” in words since it is in the beginning of a sentence.

Line 136, 138, 141, 142 : Please correct typo “Sigma-Aldritch” to “Sigma-Aldrich”.

Line 146: Please write “200 mg” in words since it is in the beginning of a sentence.

Line 178: “minimum inhibitory concentration” should be written for the first time in Line 80 “lower minimum inhibitory concentration (MIC)”.

Line 191: Please change word “antibiotic” to “antifungal”.

Line 198: Please write “100 µL” in words since it is in the beginning of a sentence.

Table 3: Why was Humulus lupulus not tested in checkerboard analysis?. In Line 222-226, it seems that H. lupulus was not excluded. It has low MIC50 (i.e., 0.625 mg/mL).

Line 236: The lowest FICI score is 0.039 (highly synergistic). Range of FICI is 0.039 to 4.0, based on data shown in Table 3.

Line 241-242: Please verify the econazole concentration as positive control. In Line 241-242: econazole concentration is 2 mg/mL, whereas in Line 191 econazole 30 µg/mL.

Figure 2: Please check again the colour in figure 2 with the data in Table 3, and consistent with the criteria of FICI (i.e., Synergistic = FICI < 0.5, additive = 0.5 ≤ FICI < 1, neutral = 1 ≤ FICI < 2, antagonistic = FICI ≥ 2). Below are some inconsistencies:

• Figure 2A. H. canadensis/P. granatum MIC90 0.50 is additive (Table 3). Therefore, the colour should be light green.

• Figure 2A. P. granatum/E. globulus MIC90 1.0 is neutral (Table 3). Therefore, the colour should be pink.

• Figure 2B. H. canadensis/P. amurense MIC90 0.50 is additive (Table 3). Therefore, the colour should be light green.

• and other cells

Figure 3: Please verify the MIC50 of H. canadensis. In table 2, MIC50 of H. canadensis is 2.5 mg/mL. However in Figure 3, in 1 mg/mL H. canadensis have >50% inhibition.

Line 348: The lowest FICI value of H. canadensis is 0.039, combined with E. globulus. Therefore, the value is “0.52 to 0.039”.

Reviewer #2: The manuscript titled “Synergistic antifungal effects of botanical extracts against Candida albicans” by Cho et al. investigated the combined action between 86 ethanolic extracts of nine plant previously reported to have strong antifungal activity on prevailing fungal pathogen C. albicans.

In this detailed study, the authors used checkerboard microdilution assays to find the synergistic, additive or antagonistic interactions. As resistance of C. albicans to available antifungals increases, the search for the alternative therapies is of high importance.

I recommend the publication of this manuscript.

Minor comments:

Please, explain the reason to choose the C. albicans strain 10231.

**Do you want your identity to be public for this peer review?** For information about this choice, including consent withdrawal, please see our Privacy Policy

Reviewer #1: No

Reviewer #2: No

---

## [Author Response · Author response to Decision Letter 1]

2 Nov 2025

Point-by-point Responses:

Revised Funding Statement:

This work was supported by NIH / ODS / NCCIH Botanical Center Grant (P50 AT002776 to IR), and the NJ Agricultural Experiment Station of Rutgers, The State University of NJ. Funding support for KA and JH was provided by the National Center for Complementary and Integrative Health through training grant 5T32AT004094. There was no additional external funding received for this study. The funders had no role in study design, data collection and analysis, decision to publish, or preparation of the manuscript.

1. Line 32: P. granatum should be written completely Punica granatum since it is stated for the first time in the manuscript.

We have changed the formatting for this as suggested.

2. Line 60: Please change word “antibiotic” to “antifungals”.

We have changed the wording as suggested.

3. Line 88: Please verify the species name Matricularia chamomilla or Matricaria chamomilla (in Table 1).

We verify that the species used for this research is Matricaria chamomilla, or German chamomile, as written in the manuscript in Table 1, and have changed the typo.

4. Table 1: In my opinion, it is better to put Table 1 in Materials and Methods rather than in Introduction.

Thank you for the suggestion, however we believe that Table 1 is best suited for the Introduction as it summarizes previous literature on bioactive compounds in the studied plants and provides context to why they were selected for the synergy testing. It also introduces important conceptual information rather than procedural information.

5. Line 97: Please write “10 µL” in words since it is in the beginning of a sentence.

Changed the structure of the sentence.

6. Line 136, 138, 141, 142 : Please correct typo “Sigma-Aldritch” to “Sigma-Aldrich”.

Changed as suggested.

7. Line 146: Please write “200 mg” in words since it is in the beginning of a sentence.

Changed the structure of the sentence.

8. Line 178: “minimum inhibitory concentration” should be written for the first time in Line 80 “lower minimum inhibitory concentration (MIC)”.

Changed as suggested.

9. Line 191: Please change word “antibiotic” to “antifungal”.

Changed as suggested.

10. Line 198: Please write “100 µL” in words since it is in the beginning of a sentence.

Changed the structure of the sentence.

11. Table 3: Why was Humulus lupulus not tested in checkerboard analysis?. In Line 222-226, it seems that H. lupulus was not excluded. It has low MIC50 (i.e., 0.625 mg/mL).

H. lupulus was not tested because our criteria for determining extracts for candidates of synergy testing was focused on higher overall percent inhibition. Although H. lupulus displayed a low MIC50, on the other hand for MIC90, it had overall poorer results. For example, although H. lupulus and P. granatum are both depicted as MIC90 ≥20, at 10 mg/mL H. lupulus has only a 50.92% inhibition whereas P. granatum has 88.10% inhibition against C. albicans. This data in Table 3 does not capture this nuance, but additional supplementary information has been added in S1_File that reflects this.

Line 236: The lowest FICI score is 0.039 (highly synergistic). Range of FICI is 0.039 to 4.0, based on data shown in Table 3.

Changed as suggested.

Line 241-242: Please verify the econazole concentration as positive control. In Line 241-242: econazole concentration is 2 mg/mL, whereas in Line 191 econazole 30 µg/mL.

Adjusted to fit most recent data collections – 30 µg/mL.

Figure 2: Please check again the colour in figure 2 with the data in Table 3, and consistent with the criteria of FICI (i.e., Synergistic = FICI < 0.5, additive = 0.5 ≤ FICI < 1, neutral = 1 ≤ FICI < 2, antagonistic = FICI ≥ 2). Below are some inconsistencies:

• Figure 2A. H. canadensis/P. granatum MIC90 0.50 is additive (Table 3). Therefore, the colour should be light green.

• Figure 2A. P. granatum/E. globulus MIC90 1.0 is neutral (Table 3). Therefore, the colour should be pink.

• Figure 2B. H. canadensis/P. amurense MIC90 0.50 is additive (Table 3). Therefore, the colour should be light green.

• and other cells

Changed the ranges on Figure 2 so values at the thresholds (i.e. 0.500, 1.000, etc.) have appropriate colors for their synergy designation.

Figure 3: Please verify the MIC50 of H. canadensis. In table 2, MIC50 of H. canadensis is 2.5 mg/mL. However in Figure 3, in 1 mg/mL H. canadensis have >50% inhibition.

MIC50 is 2.5 mg/mL. The original value of % inhibition at 1 mg/mL in Figure 3 was a mistake and has been fixed.

Line 348: The lowest FICI value of H. canadensis is 0.039, combined with E. globulus. Therefore, the value is “0.52 to 0.039”.

Changed as suggested.

Minor comments:

Please, explain the reason to choose the C. albicans strain 10231.

We chose to use C. albicans strain 10231 because it is widely studied and characterized and is a designated strain for QC standards for various assays by CLSI and EUCAST, whose methods were used to base our own study methodology on.

---

## [Editor Report · Decision Letter 1]

5 Nov 2025

*Candida albicans*

Dear Dr. Cho,

Thank you for submitting your manuscript to PLOS ONE. After careful consideration, we feel that it has merit but does not fully meet PLOS ONE’s publication criteria as it currently stands. Therefore, we invite you to submit a revised version of the manuscript that addresses the points raised during the review process.

We look forward to receiving your revised manuscript.

Kind regards,

Theerapong Krajaejun, M.D.

Academic Editor

PLOS ONE

Journal Requirements:

Additional Editor Comments:

- Regarding Table 1, while your intention to summarize prior literature and provide conceptual context is noted, relocation to the Materials and Methods section is recommended, consistent with the reviewer’s suggestion and standard scientific conventions. The Introduction should remain a narrative overview supported by citations, rather than a place for structured data. Placing the table in the Methods section, where the selection of plant species is described, will improve clarity, enhance manuscript flow, and allow for more appropriate referencing in the Results and Discussion when interpreting outcomes related to the extracts.

- Please include a brief rationale in the Methods section for selecting Candida albicans strain ATCC 10231. This will help clarify its relevance and support reproducibility.

---

## [Author Response · Author response to Decision Letter 2]

20 Dec 2025

We thank the Editor and the Reviewers for their careful evaluation of our manuscript and for providing insightful comments that have helped us improve the quality and clarity of our work. Below, we provide a detailed, point-by-point response to all comments. Changes made in the revised manuscript are indicated by tracked changes.

Point-by-point Responses:

1. Regarding Table 1, while your intention to summarize prior literature and provide conceptual context is noted, relocation to the Materials and Methods section is recommended, consistent with the reviewer’s suggestion and standard scientific conventions. The Introduction should remain a narrative overview supported by citations, rather than a place for structured data. Placing the table in the Methods section, where the selection of plant species is described, will improve clarity, enhance manuscript flow, and allow for more appropriate referencing in the Results and Discussion when interpreting outcomes related to the extracts.

Moved Table 1 to the “Plant materials” subsection in Materials and Methods, as recommended, with additional brief context on their selection.

2. Please include a brief rationale in the Methods section for selecting Candida albicans strain ATCC 10231. This will help clarify its relevance and support reproducibility.

Provided a brief explanation of ATCC 10231 in “Reagents, chemicals and fungal strain” subsection in Materials and Methods as recommended.

---

## [Editor Report · Decision Letter 2]

26 Dec 2025

Synergistic antifungal effects of botanical extracts against *Candida albicans*

PONE-D-25-38285R2

Dear Dr. Cho,

We’re pleased to inform you that your manuscript has been judged scientifically suitable for publication and will be formally accepted for publication once it meets all outstanding technical requirements.

Kind regards,

Theerapong Krajaejun, M.D.

Academic Editor

PLOS One
---

## [Editor Report · Acceptance letter]

PONE-D-25-38285R2

PLOS One

Dear Dr. Cho,

I'm pleased to inform you that your manuscript has been deemed suitable for publication in PLOS One. Congratulations! Your manuscript is now being handed over to our production team.

Kind regards,

on behalf of

Dr. Theerapong Krajaejun

Academic Editor

PLOS One